# AudioMosaic: Contrastive Masked Audio Representation Learning

**Hanxun Huang** [1]  **Qizhou Wang** [1]  **Xingjun Ma** [2]  **Cihang Xie** [3]  **Christopher Leckie** [1]  **Sarah Erfani** [1]

## Abstract

Audio self-supervised learning (SSL) aims to learn general-purpose representations from large-scale unlabeled audio data. While recent advances have been driven mainly by generative reconstruction objectives, contrastive approaches remain less explored, partly due to the difficulty of designing effective audio augmentations and the large batch sizes required for contrastive pre-training. We introduce **AudioMosaic**, a contrastive learning–based audio encoder for general audio understanding. During pre-training, AudioMosaic constructs positive pairs by applying structured time–frequency masking to spectrogram patches, which reduces memory usage and enables efficient large-batch training. Compared with generative approaches, the AudioMosaic encoder learns more discriminative utterance-level representations that demonstrate strong transferability across datasets, domains, and acoustic conditions. Extensive experiments show that AudioMosaic achieves state-of-the-art performance on several standard audio benchmarks under both linear probing and fine-tuning. We further show that integrating the pretrained AudioMosaic encoder into audio–language models improves performance on audio–language tasks. The code is publicly available in our GitHub repository.

## 1. Introduction

Self-supervised learning (SSL) has become a cornerstone of representation learning across modalities, driving advances in natural language processing (Radford et al.; Devlin et al., 2019), computer vision (Chen et al., 2020c; He et al., 2022;

Oquab et al., 2024; Fan et al., 2025), and audio processing (Baevski et al., 2020; Fei et al., 2023; Ahmed et al., 2024; Kong et al., 2024; Lee et al., 2025; Ghosh et al., 2025; Goel et al., 2025; Dong et al., 2025). Existing audio SSL methods largely fall into two paradigms: masked modeling and contrastive learning. Masked modeling approaches, particularly masked spectrogram modeling, have been extensively explored for general audio understanding (Niizumi et al., 2022; Huang et al., 2022; Chong et al., 2023; Chen et al., 2024; Alex et al., 2025). In contrast, contrastive learning has primarily focused on raw waveform inputs and speech-centric tasks (Oord et al., 2018; Baevski et al., 2020; Hsu et al., 2021). Despite its success in vision and language, contrastive learning over spectrogram representations for audio understanding remains relatively underexplored.

This gap is not due to a lack of effort, but reflects fundamental challenges in applying contrastive learning to spectrograms. Effective contrastive learning relies on carefully designed data augmentations to generate informative positive pairs (Wang et al., 2022; Zhai et al., 2024). However, augmentation design is highly domain-dependent (Blankemeier et al., 2023; Zhou et al., 2024) and often requires expensive search over combinations of transformations (Chen et al., 2020c). Moreover, contrastive methods typically rely on large batch sizes to provide sufficient negative samples, leading to substantial computational cost. Although SpecAugment (Park et al., 2019) is highly effective for supervised learning, it has been shown to be suboptimal for self-supervised masked modeling objectives (Huang et al., 2022). Together, these observations underscore the difficulty of directly applying existing methods to spectrogram.

In this work, we introduce **AudioMosaic**, an audio encoder pre-trained with a contrastive objective and a tailored augmentation strategy inspired by SpecAugment. Unlike SpecAugment, which directly masks spectrogram regions with zero values for supervised learning, AudioMosaic operates on spectrogram patches and constructs positive pairs by applying independent masking along the time and frequency dimensions. This produces complementary views of the same utterance, as illustrated in Figure 1(a). In contrast to the unstructured random masking used in reconstruction-based models to capture local correlations, the structured masking in AudioMosaic is designed specifically for the contrastive setting to encourage the learning of global, dis-

[1]School of Computing and Information Systems, The University of Melbourne, Australia [2]Institute of Trustworthy Embodied AI, Fudan University, China [3]Baskin School of Engineering, University of California, Santa Cruz, USA. Correspondence to: Hanxun Huang <hanxun@unimelb.edu.au>, Christopher Leckie <caleckie@unimelb.edu.au>.

*Proceedings of the $43^{rd}$ International Conference on Machine Learning*, Seoul, South Korea. PMLR 306, 2026. Copyright 2026 by the author(s).

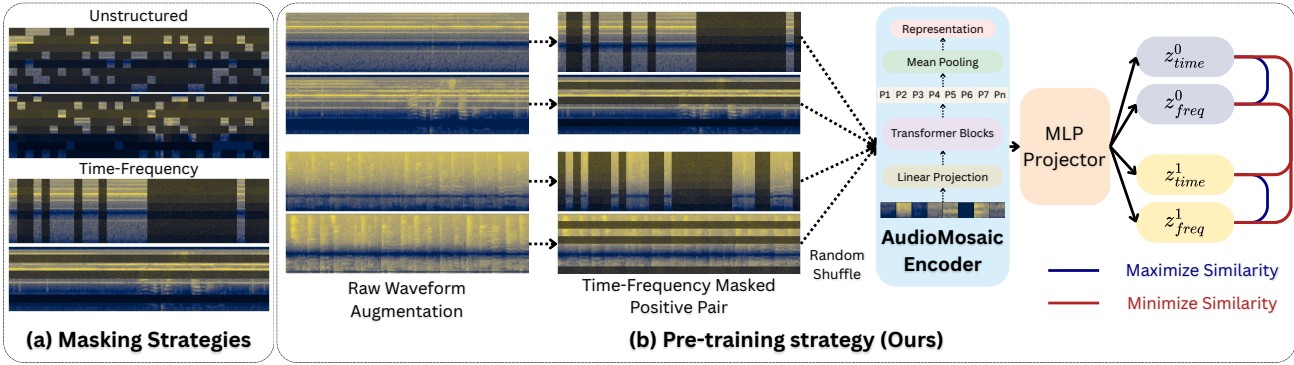

Figure 1. Overview of AudioMosaic. *(a)* Comparison between commonly used unstructured masking in prior audio-SSL methods and the time–frequency masking strategy. *(b)* AudioMosaic first converts an input spectrogram into patch tokens with positional encoding, then applies time–frequency masking to construct a positive pair for contrastive learning. The masked patches are omitted, and only the visible ones are fed into the encoder in randomized order.

criminative representations. During pre-training, only visible patches are processed by the encoder, and contrastive learning is performed across masked views within a shared embedding space. An overview of the framework is shown in Figure 1(b). This time–frequency masking strategy differs from prior waveform-based contrastive methods (Oord et al., 2018; Baevski et al., 2020), which focus on contrasting short-term temporal views, and from masked spectrogram modeling approaches (Huang et al., 2022), which rely on local context for reconstruction.

We further analyze why this augmentation strategy is more effective. Spectrograms exhibit strong local correlations along both time and frequency dimensions. For reconstruction-based objectives, unstructured masking that preserves local structure is beneficial because accurate reconstruction relies heavily on local context. In contrastive learning, however, if positive views share too much local structure, the contrastive task becomes overly easy and fails to encourage the learning of informative features. This can lead to dimension collapse (Jing et al., 2022; Huang et al., 2024). Such collapse is characterized by low effective rank (Roy & Vetterli, 2007) and degraded representation quality. Structured time–frequency masking alleviates this issue by reducing shared local redundancy between positive views while preserving broader temporal–spectral patterns. By contrasting complementary time–frequency masked views of the same utterance, the model is encouraged to rely on more global structure, leading to the learning of utterance-invariant representations that are more discriminative and transferable across domains.

Extensive experiments on standard benchmarks, including AudioSet (Gemmeke et al., 2017), ESC-50 (Piczak, 2015), Speech Commands (Warden, 2018), and Environmental Sound Deepfake Detection (EnvSDD) (Yin et al., 2025a;b), show that AudioMosaic achieves state-of-the-art performance on several tasks, while reducing pre-training

memory cost and model complexity. In addition, following Gong et al. (2024), we show that aligning the AudioMosaic encoder with large language models further improves performance on audio–language tasks.

In summary, our main contributions are as follows:

- We introduce **AudioMosaic**, a contrastive audio pre-training framework that rethinks masking as a mechanism for constructing informative positive pairs rather than as noise for reconstruction. By using structured, independent time–frequency masking, AudioMosaic creates complementary but non-trivial views that enable effective contrastive learning on spectrograms while remaining computationally efficient.

- We provide a representation-level analysis of masking strategies using effective rank, offering insight into how excessive shared local structure in positive pairs can lead to degenerate contrastive solutions. This analysis helps explain why structured time–frequency masking is particularly effective for contrastive learning on spectrograms.

- We demonstrate that AudioMosaic learns discriminative utterance-level representations that generalize across datasets, domains, and acoustic conditions, achieving state-of-the-art results on several standard benchmarks and strong performance on others, while also supporting memory efficient large-batch pre-training and improving audio–language models.

## 2. Relate Work

**Masked Modeling.** Masked modeling has emerged as a popular paradigm for self-supervised pre-training due to its simplicity and effectiveness: by reconstructing masked or missing content, models can learn context-aware representations. The concept was first popularized by masked

language modeling (Devlin et al., 2019), and subsequently adapted by the computer vision community in masked image modeling (Chen et al., 2020b; Feichtenhofer et al., 2022; Wei et al., 2022; 2023; Tong et al., 2022; Xie et al., 2023; Huang et al., 2023). For instance, the MAE (He et al., 2022) proposed reconstructing masked image patches with a continuous regression objective, while BEiT (Bao et al., 2022) predicted discrete visual tokens generated by a pretrained VAE (Ramesh et al., 2021). Recent works have extended this paradigm to masked spectrogram modeling for audio self-supervised learning (Niizumi et al., 2022; Baade et al., 2022; Chong et al., 2023). Audio-MAE (Huang et al., 2022) adapted the MAE framework to the audio domain and systematically studied different masking strategies, finding that unstructured random masking outperforms structured time–frequency masking. Similarly, BEATs (Chen et al., 2023) adopted a discrete token prediction objective, analogous to BEiT, by learning to predict masked acoustic tokens. The masked spectrogram modeling paradigm has also been extended to teacher–student distillation frameworks (Chen et al., 2024; Alex et al., 2025). While reconstruction-based objectives encourage models to capture fine-grained local correlations within a spectrogram, they inherently rely on neighboring visible regions to infer the masked content.

**Contrastive Learning.** Contrastive learning is another popular framework for self-supervised representation learning by encouraging embeddings of different augmented views of the same sample (positives) to be similar, while pushing apart those from different samples (negatives). This paradigm has achieved remarkable success in computer vision (Oord et al., 2018; He et al., 2020; Chen et al., 2020c; Grill et al., 2020; Bardes et al., 2022) and multi-modal learning (Radford et al., 2021; Elizalde et al., 2023; Gong et al., 2023; Jenni et al., 2023). In the audio domain, prior works have focused on contrastive learning over latent representations of masked audio segments to learn general-purpose speech representations (Oord et al., 2018; Baevski et al., 2020). These methods typically operate on raw waveforms or intermediate feature sequences, emphasizing temporal continuity as a primary source of self-supervision. For spectrogram-based learning, COLA (Saeed et al., 2021) proposed using segments from the same clip as positives and those from different clips as negatives. BYOL-A (Niizumi et al., 2021) extended this idea using augmentation and bootstrapping. SSAST (Gong et al., 2022a) further explored patch-level contrastive objectives within masked spectrogram modeling to jointly learn discriminative and reconstructive representations.

In contrast to prior masked modeling and contrastive learning approaches, our method performs *utterance-level contrastive pre-training on spectrograms, using masking as a view-generation mechanism*. By contrasting complementary time–frequency masked views of the same utterance, the model learns global, utterance-invariant representations.

**Out-of-domain Pre-training for Audio.** Transferring ImageNet-supervised pre-trained models (Deng et al., 2009; He et al., 2016; Tan & Le, 2019; Dosovitskiy et al., 2021; Touvron et al., 2021) has become a common practice in audio representation learning (Gong et al., 2021; Nagrani et al., 2021; Koutini et al., 2022; Chen et al., 2022). These models are typically adapted to operate on audio spectrograms by modifying the input layer from three RGB channels to a single-channel spectrogram input. In contrast, our method avoids reliance on out-of-domain (non-audio) supervision and instead focuses on audio-only self-supervised pre-training from scratch.

## 3. AudioMosaic

In this section, we introduce the AudioMosaic encoder. Our goal is to learn generalizable audio representations that transfer across downstream tasks and conditions.

**Time–Frequency Masking for Positive Pair Construction.** Given a raw waveform input $r$, we apply simple temporal and acoustic augmentations to obtain two views $r_1$ and $r_2$ of the same instance. These augmentations are essential to prevent the two views from containing identical or highly similar patches, which could otherwise lead the model to learn trivial representations. Each view is then transformed into a log-Mel spectrogram $\mathbf{x} \in \mathbb{R}^{t \times f} = \mathcal{T}_{\mathrm{mel}}(r)$, where $\mathcal{T}_{\mathrm{mel}}(\cdot)$ denotes the log-Mel transformation operator, and $t$ and $f$ represent the number of time frames and Mel-frequency bins, respectively. Each spectrogram $\boldsymbol{x}_i = \mathcal{T}_{\mathrm{mel}}(r_i)$ is partitioned into patches of size $p_t \times p_f$, forming a sequence of $N = \frac{t}{p_t} \times \frac{f}{p_f}$ patch embeddings $\mathbf{h} \in \mathbb{R}^{N \times d}$. We apply masking independently along the time and frequency dimensions to two augmented views ($r_1$ and $r_2$) of the same utterance. Let $M_t(\cdot)$ and $M_f(\cdot)$ denote masking operators that randomly drop patches in contiguous time regions and along frequency bands, respectively. The masked patch sequences are given by

$$\mathbf{h}_t = M_t(\mathbf{h}_1), \quad \mathbf{h}_f = M_f(\mathbf{h}_2). \tag{1}$$

Each masking operator is controlled by a masking ratio parameter, $\rho_t$ and $\rho_f$, which determine the proportion of time and frequency patches removed from each view.

**Contrastive Pre-training.** Each masked patch sequence, $\mathbf{h}_t$ and $\mathbf{h}_f$, is first augmented with 2D positional embeddings to preserve the spatial structure of the time–frequency patches. To enhance invariance and reduce spatial bias, the order of patch tokens is randomly shuffled before encoding. The resulting sequences are then processed by a shared Transformer encoder $f_\theta(\cdot)$, parameterized by $\theta$, to obtain latent representations:

$$\mathbf{q}_t = f_\theta(\mathbf{h}_t), \quad \mathbf{q}_f = f_\theta(\mathbf{h}_f). \tag{2}$$

A lightweight projection MLP head $g_\phi(\cdot)$ is applied to map these representations onto a normalized embedding space:

$$\mathbf{z}_t = g_\phi(\mathbf{q}_t), \quad \mathbf{z}_f = g_\phi(\mathbf{q}_f), \tag{3}$$

where $\mathbf{z}_t, \mathbf{z}_f \in \mathbb{R}^d$ are $\ell_2$-normalized embeddings. The model is trained to maximize the agreement between complementary time–frequency masked views of the same utterance while minimizing similarity to other samples in the batch. We adopt the standard contrastive loss defined as:

$$\mathcal{L} = -\frac{1}{2B} \sum_{i=1}^{B} \left[ \log \frac{\exp(\text{sim}(\mathbf{z}_t^{(i)}, \mathbf{z}_f^{(i)})/\tau)}{\sum_{j=1}^{B} \exp(\text{sim}(\mathbf{z}_t^{(i)}, \mathbf{z}_f^{(j)})/\tau)} \right.$$
$$\left. + \log \frac{\exp(\text{sim}(\mathbf{z}_f^{(i)}, \mathbf{z}_t^{(i)})/\tau)}{\sum_{j=1}^{B} \exp(\text{sim}(\mathbf{z}_f^{(i)}, \mathbf{z}_t^{(j)})/\tau)} \right], \tag{4}$$

where $\text{sim}(\cdot, \cdot)$ denotes cosine similarity, $\tau$ is temperature parameter, and $B$ is the batch size.

**Downstream Tasks.** After pre-training, we retain only the encoder and discard the MLP projection head, following prior work (Huang et al., 2022; Chen et al., 2020c). The projection head is replaced with a task-specific linear classifier for audio classification tasks, or used to align the encoder with a LLM for multimodal audio–language tasks.

**Efficiency.** Unlike prior work (Huang et al., 2022), which relies on a Transformer-based decoder to reconstruct the spectrogram, AudioMosaic uses a lightweight MLP projection head. This projection head introduces only a small number of additional parameters, making the overall framework substantially more parameter- and memory-efficient during pre-training.

In addition, higher masking ratios not only encourage stronger invariance to missing temporal or spectral information, but also reduce the number of visible tokens, thereby improving training efficiency. Specifically, masking reduces the Transformer's quadratic attention cost from $O(N^2)$ to $O((1-\rho)^2 N^2)$, where $N$ is the total number of patches and $\rho$ denotes the overall masking ratio (e.g., masking 50% of the patches yields a 75% reduction in quadratic attention computation). The reduced token count also lowers memory consumption, enabling substantially larger batch sizes, which is beneficial for effective contrastive pre-training.

## 4. Analysis of Masking Strategies

We first review the preliminaries needed to analyze how different masking strategies influence representation quality.

**Effective Rank.** We follow the standard definition of effective rank (Roy & Vetterli, 2007) based on the entropy of the singular values. Let $A \in \mathbb{C}^{M \times N}$ be a non-zero matrix with singular values:

$$\sigma_1 \geq \sigma_2 \geq \cdots \geq \sigma_Q \geq 0, \qquad Q = \min\{M, N\}.$$

Define the singular value distribution:

$$p_k = \frac{\sigma_k}{\sum_{j=1}^{Q} \sigma_j}, \quad k = 1, \ldots, Q.$$

The effective rank of $A$ is given by

$$\text{erank}(A) = \exp\left(-\sum_{k=1}^{Q} p_k \log p_k\right),$$

which corresponds to the exponential of the Shannon entropy of the normalized singular values.

**Representation Quality.** Effective rank is a well-established measure for assessing representation quality without relying on downstream fine-tuning or linear probing. It can be interpreted as an estimate of the global intrinsic dimensionality (Pettis et al., 1979; Bruske & Sommer, 2002) of learned representations. Prior work has shown that effective rank correlates with downstream performance (Dubois et al., 2023; Garrido et al., 2023), and that higher intrinsic dimensionality is generally associated with richer and more expressive representations (Zhang et al., 2022; Huang et al., 2024). Conversely, extremely low intrinsic dimensionality often indicates degenerate solutions. This includes dimension collapse (Jing et al., 2022), where representations lie in a low-dimensional subspace, and mode collapse, an extreme case in which all representations converge to a single vector. Both phenomena degrade representation quality. We do not claim that higher intrinsic dimensionality necessarily implies better generalization; rather, we use effective rank as a diagnostic signal, where unusually low intrinsic dimensionality suggests poor or collapsed representations, consistent with prior work. Therefore, effective rank provides a useful tool for evaluating and designing SSL methods.

For each encoder, we extract representations of the form $\mathbf{q} = f_\theta(M(\mathbf{h}))$, where $\mathbf{q} \in \mathbb{R}^d$, $f_\theta$ denotes the encoder, and $M$ is the masking operator applied at inference time. The encoder is trained either with Audio-MAE (Huang et al., 2022) or with the contrastive loss in Eq. (4). The masking operator used at inference may match or differ from the one used during training. To estimate effective rank, we collect a batch of representations and form a matrix $Z \in \mathbb{R}^{B \times d}$, where $B$ is the batch size. We compute the singular values of $Z$ and use them to estimate the effective rank, which serves as a proxy for the intrinsic dimensionality of the representation set. Since each instance captures different content, higher effective rank indicates that representations span a richer subspace, whereas very low effective rank suggests degenerate or collapsed representations.

We compare a contrastive objective with the generative reconstruction objective used in Audio-MAE (Huang et al., 2022), which employs unstructured masking during pre-training. We additionally vary the masking strategy used to

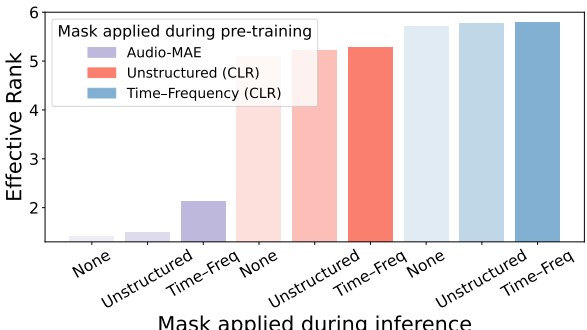

*Figure 2.* Comparison of the effective rank of encoder representations. Encoders are trained with either unstructured masking or the proposed time–frequency masking, using identical hyperparameters and masking ratios. "Inference-time masking" denotes the masking strategy applied when extracting representations. "None" indicates that no masking is applied during inference.

form contrastive views (unstructured vs. time–frequency), resulting in three encoders: (i) Audio-MAE (reconstruction + unstructured masking), (ii) contrastive pre-training with unstructured masking, and (iii) contrastive pre-training with time–frequency masking (AudioMosaic). For each encoder, we extract representations under three inference-time settings: no masking ("none"), unstructured masking, and time–frequency masking. The resulting effective ranks are shown in Figure 2.

Encoders trained with contrastive learning consistently achieve higher effective rank than Audio-MAE, across inference-time masking choices. Time–frequency masking at inference yields a higher effective rank than no masking and unstructured masking for all encoders (solid vs. transparent bars), suggesting that structured masking produces representations that better utilize the embedding space.

For contrastive pre-training in particular, time–frequency masking leads to the highest effective rank, indicating that the learned representations occupy a higher-dimensional subspace and are less prone to degenerate solutions. In contrast, unstructured masking yields lower effective rank, consistent with representations that concentrate in fewer directions. Notably, AudioMosaic (blue bars) achieves the highest effective rank among all methods.

Overall, these results suggest that structured time–frequency masking for constructing positive pairs is associated with richer representations, in line with prior analyses of representation quality.

## 5. Experiments

The goal of audio SSL is to learn generalizable representations that improve performance across diverse downstream datasets and tasks. In the following experiments, we com-

prehensively evaluate AudioMosaic on a range of benchmarks to assess its generalization and transfer performance. All experiments are conducted on NVIDIA L40S GPUs (48GB). We use publicly available pre-trained checkpoints for baseline methods and closely follow their original hyperparameters and codebases.

**Datasets and Metrics.** Following prior work (Huang et al., 2022; Chen et al., 2023; 2024; Alex et al., 2025), we evaluate on widely used audio benchmarks. For pre-training, we use AudioSet (Gemmeke et al., 2017) without labels, combining the unbalanced and balanced splits. Due to distribution constraints, we only download 1.91M samples from the unbalanced split, 20k from the balanced split, and 19k from the evaluation split. We follow standard practice and pre-train on both unbalanced and balanced splits. All audio is resampled to mono at 16 kHz and converted into log Mel-spectrograms with 128 Kaldi-compatible Mel bands (Povey et al., 2011), using a 25 ms Hann window and 10 ms hop size. A 10-second clip yields a spectrogram of size $1 \times 1024 \times 128$.

For downstream evaluation, we fine-tune on AS-2M (unbalanced) and AS-20k (balanced), using the same weighted sampling strategy for AS-2M as in Huang et al. (2022). We further evaluate on ESC-50 (Piczak, 2015) and Speech Commands (SPC-1, SPC-2) (Warden, 2018). We report mean average precision (mAP) on AS-2M and AS-20k, and classification accuracy on ESC-50, SPC-1, and SPC-2.

**Audio–LLM Alignment.** For audio–language evaluation, we follow the LTU setup (Gong et al., 2024), aligning an audio encoder with LLaMA-7B (Touvron et al., 2023) using curriculum training. Due to distribution constraints, we use 5.4M of the original 5.6M samples from OpenAQA. Evaluation follows the LTU protocol and includes audio classification on Vocal Sound (Gong et al., 2022b), TUT Acoustic Scenes (Mesaros et al., 2018), Beijing Opera Percussion Instrument (BJO) (Tian et al., 2014), DCASE (Kong et al., 2020b), VGGSound (Chen et al., 2020a), FSD-50K (Fonseca et al., 2021), ESC-50, and AudioSet. We also evaluate audio captioning on Clotho (Drossos et al., 2020) and AudioCaps (Kim et al., 2019). We report micro F1-score on DCASE and SPICE (Anderson et al., 2016) for captioning.

**Architectures and Pre-Training Details.** We adopt the Audio Spectrogram Transformer (AST) (Gong et al., 2021), which applies a ViT (Dosovitskiy et al., 2021) encoder directly to spectrograms. We use a 12-layer ViT-B/16 with $16 \times 16$ patches. The projection head is a two-layer MLP: a linear layer with 512 hidden units, Batch Normalization (Ioffe & Szegedy, 2015), ReLU, followed by a linear layer to the projection dimension, and a final bias-free Batch-Norm. The projection dimension is 128. We use AdamW (Loshchilov & Hutter, 2019) with learning rate $6 \times 10^{-4}$ and weight decay 0.01. The masking ratios are set to $\rho_t = 0.6$

*Table 1.* Comparison with state-of-the-art methods on audio and speech downstream tasks. *PT*: pre-training, *FT*: fine-tuning, *AS*: AudioSet, *LS*: LibriSpeech (Panayotov et al., 2015) and *IN*: ImageNet (Deng et al., 2009). *TI* and *CLAP* duse audio–text paired datasets from multiple sources.*linear evaluation results from Yang et al. (2021). Best results are shown in **bold**.

| Model | Backbone | Data (PT) | Params (PT) | Params (FT) | AS-20K | Audio AS-2M | ESC-50 | Speech SPC-2 | SPC-1 |
|---|---|---|---|---|---|---|---|---|---|
| **No pre-training** | | | | | | | | | |
| PANN (Kong et al., 2020a) | CNN | - | - | 81M | 27.8 | 43.1 | 83.3 | 61.8 | - |
| ERANN (Verbitskiy et al., 2022) | CNN | - | - | 57M | - | 45.0 | 89.2 | - | - |
| **Out-of-domain supervised pre-training** | | | | | | | | | |
| AST (Gong et al., 2021) | ViT-B/16 | IN | 86M | 86M | 34.7 | 45.9 | 88.7 | 98.1 | 95.5 |
| MBT (Nagrani et al., 2021) | ViT-B/16 | IN-21K | 343M | 86M | 31.3 | 44.3 | - | - | - |
| **In-domain language supervised pre-training** | | | | | | | | | |
| Wav2CLIP (Wu et al., 2022) | ResNet | TI+AS | 74M | 74M | - | - | 86.0 | - | - |
| AudioCLIP (Guzhov et al., 2022) | ESResNeXt | TI+AS | 134M | 93M | - | 25.9 | 96.7 | - | - |
| CLAP (Elizalde et al., 2023) | HTS-AT | CLAP | 159M | 159M | - | - | 91.0 | - | - |
| **In-domain self-supervised pre-training** | | | | | | | | | |
| Wav2Vec 2.0 (Baevski et al., 2020) | ViT-B/16 | LS | 95M | 95M | - | - | - | - | 96.2* |
| HuBERT (Hsu et al., 2021) | ViT-B/16 | LS | 95M | 95M | - | - | - | - | 96.3* |
| SS-AST (Gong et al., 2022a) | ViT-B/16 | AS+LS | 89M | 89M | 31.0 | - | 88.8 | 98.0 | 96.0 |
| MAE-AST (Baade et al., 2022) | ViT-B/16 | AS+LS | 174M | 86M | 30.6 | - | 90.0 | 97.9 | 95.8 |
| COLA (Saeed et al., 2021) | CNN | AS | 5M | 5M | - | - | - | 76.8 | 76.7 |
| BYOL-A (Niizumi et al., 2021) | CNN | AS | 5M | 5M | - | - | - | 92.2 | 91.0 |
| Conformer-SSL (Srivastava et al., 2022) | Conformer | AS | 88M | 88M | - | 41.1 | 88.0 | - | - |
| Data2Vec 2.0 (Baevski et al., 2022) | ViT-B/16 | AS | 94M | 94M | 34.5 | - | - | - | - |
| MSM-MAE (Niizumi et al., 2022) | ViT-B/16-8 | AS | 93M | 86M | - | - | 85.6 | 87.3 | - |
| Audio-MAE (Huang et al., 2022) | ViT-B/16 | AS | 137M | 86M | 37.0 | 47.3 | 94.1 | 98.3 | 96.9 |
| MaskSpec (Chong et al., 2023) | ViT-B/16 | AS | 112M | 86M | 32.3 | 47.1 | 89.6 | 97.7 | - |
| BEATs$_{iter3}$ (Chen et al., 2023) | ViT-B/16 | AS | 182M | 91M | 38.3 | 48.0 | 95.6 | 98.3 | 97.7 |
| A-JEPA (Fei et al., 2023) | ViT-B/16 | AS | 354M | 86M | 38.4 | 48.6 | 96.3 | 98.5 | 97.7 |
| ASiT (Ahmed et al., 2024) | ViT-B/16 | AS | 96M | 86M | 37.4 | 47.5 | 94.2 | **98.8** | 98.2 |
| EAT (Chen et al., 2024) | ViT-B/16 | AS | 93M | 88M | 40.2 | 48.6 | 95.9 | 98.3 | - |
| SSLAM (Alex et al., 2025) | ViT-B/16 | AS | 93M | 88M | 40.9 | **50.2** | 96.2 | 98.1 | 98.8 |
| AudioMosaic (Ours) | ViT-B/16 | AS | 86M | 86M | **42.5** | 50.2 | **97.5** | 98.4 | **99.0** |

and $\rho_f = 0.4$. Full hyperparameters are in Appendix A.

**Baselines.** We primarily compare against recent in-domain self-supervised methods, including Audio-MAE (Huang et al., 2022), BEAT (Chen et al., 2023), EAT (Chen et al., 2024), and SSLAM (Alex et al., 2025). We also include contrastive baselines, namely COLA (Saeed et al., 2021) and BYOL-A (Niizumi et al., 2021). For completeness, we additionally report results for in-domain supervised pre-training, out-of-domain self-supervised pre-training, and training from scratch. All baseline fine-tuning results are taken from the values reported in the original papers.

### 5.1. Fine-tuning Evaluation

We compare AudioMosaic with prior methods on standard benchmarks, where all models are fine-tuned from a pre-trained encoder on a diverse set of audio and speech downstream tasks. Results are shown in Table 1. AudioMosaic achieves state-of-the-art performance on several benchmarks and remains competitive on the others, consis-

tently outperforming masked spectrogram modeling methods (Audio-MAE, MaskSpec) and their enhanced variants (BEATs, EAT, SSLAM).

On AS-20K, AudioMosaic achieves 42.5 mAP, surpassing the strongest baseline, SSLAM (40.9 mAP), demonstrating the benefit of improved pre-training when labeled data are limited. On AS-2M, AudioMosaic matches the performance of SSLAM. Since AS-2M is used for both pre-training and fine-tuning, performance is near saturation and thus less sensitive to differences in representation quality. Beyond AudioSet, AudioMosaic also improves performance on ESC-50 and SPC, indicating strong transferability across different domains, tasks, and acoustic conditions.

The results further suggest that in-distribution audio self-supervised pre-training is more effective for general-purpose audio representations than language-supervised or out-of-domain pre-training. In addition, AudioMosaic is computationally efficient: most parameters (about 86M) belong to the backbone encoder, while the projection head adds negli-

*Table 2.* Linear probing evaluation is performed with a frozen encoder. Results for baseline models are obtained using their officially released weights. All evaluation metrics follow Table 1. Best results are shown in **bold**.

| Model | AS-20K | AS-2M | ESC-50 |
|---|---|---|---|
| Audio-MAE (Huang et al., 2022) | 18.3 | 20.5 | 86.9 |
| BEATs$_{iter3}$ (Chen et al., 2023) | 8.2 | 12.2 | 72.7 |
| EAT (Chen et al., 2024) | 12.5 | 18.4 | 83.5 |
| SSLAM (Alex et al., 2025) | 15.0 | 19.5 | 87.1 |
| AudioMosaic (Ours) | **29.4** | **28.7** | **93.0** |

gible overhead. In contrast, masked spectrogram modeling requires an additional Transformer-based decoder, nearly doubling the parameter count during pre-training.

### 5.2. Linear Probing Evaluation

We additionally evaluate representation quality using linear probing, where the encoder is frozen and only a linear classifier is trained. This setting isolates the quality of learned features without adaptation and has been shown to correlate well with transfer performance (Chen et al., 2020c; Ericsson et al., 2021; Marks et al., 2025). We use the average-pooled output of the last encoder layer over the token sequence as the representation. Results are in Table 2.

AudioMosaic substantially outperforms masked spectrogram modeling methods under linear probing. Notably, strong fine-tuning performance does not necessarily imply strong linear probing performance: although BEATs, EAT, and SSLAM improve over Audio-MAE under fine-tuning, they do not consistently outperform Audio-MAE under linear probing. This observation is consistent with recent works (Rauch et al., 2025; Psomas et al., 2026) in both vision and audio showing that fine-tuning can obscure differences in representation quality, and that standard probing protocols may fail to faithfully reflect the utility of pre-trained embeddings.

Recent works (Yang et al., 2021; Alkin et al., 2025; Liang et al., 2025; Huang et al., 2025) have also suggested that representations from different layers can yield substantially different downstream performance. Here, we further examine this effect by conducting linear probing experiments using representations from different layers, as well as weighted-sum and attention-based combinations of representations across layers. Results are in Figure 3. AudioMosaic representations improve monotonically with depth, peaking at layer 10 (30.2 mAP) with minimal degradation at the final layer, indicating that deeper layers consistently encode richer semantic content. In contrast, EAT, BEATs, and SSLAM peak at middle layers (layers 5–8) and degrade sharply at layer 11, suggesting that their final layers over-specialize toward pretraining objectives at the expense of transferability. AudioMAE improves steadily but remains

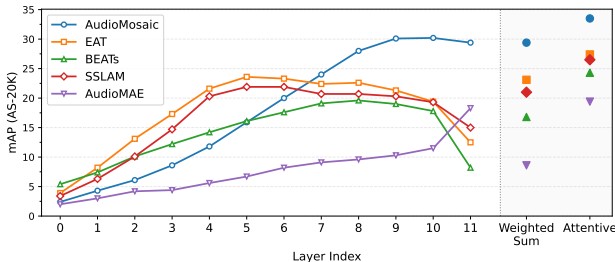

*Figure 3.* Layer-wise linear probe results on AudioSet-20K (mAP). Each layer's output is frozen and a linear classifier is trained on top. *Weighted Sum* aggregates all layers with learned weights; *Attentive* uses a learned attention query over all layer representations.

*Table 3.* Deepfake detection performance on the EnvSDD dataset (Yin et al., 2025b), reported in Equal Error Rate (EER, %). "Seen SD" denotes seen source datasets, and "Seen GM" denotes seen generative models. Results are reported separately for text-to-audio (TTA) and audio-to-audio (ATA) deepfakes. Best results are shown in **bold**.

| System | Test Set | Test Condition | | Fake Type | |
|---|---|---|---|---|---|
| | | Seen SD | Seen GM | TTA | ATA |
| Wav2Vec 2.0 +AASIST (Yin et al., 2025b) | Test 01 | ✓ | ✓ | 0.26 | 0.38 |
| | Test 02 | ✓ | ✗ | 13.04 | 26.59 |
| | Test 03 | ✗ | ✓ | 10.60 | 13.30 |
| | Test 04 | ✗ | ✗ | 45.80 | 52.40 |
| | Average | - | - | 17.43 | 23.17 |
| BEATs +AASIST (Yin et al., 2025b) | Test 01 | ✓ | ✓ | 0.08 | 0.03 |
| | Test 02 | ✓ | ✗ | 1.26 | 0.08 |
| | Test 03 | ✗ | ✓ | 4.70 | 2.20 |
| | Test 04 | ✗ | ✗ | 17.20 | 3.00 |
| | Average | - | - | 5.81 | 1.33 |
| AudioMosaic +Linear (Ours) | Test 01 | ✓ | ✓ | 0.00 | 0.00 |
| | Test 02 | ✓ | ✗ | 0.05 | 0.00 |
| | Test 03 | ✗ | ✓ | 0.38 | 0.03 |
| | Test 04 | ✗ | ✗ | 4.80 | 0.03 |
| | Average | - | - | **1.30** | 0.02 |
| AudioMosaic +AASIST (Ours) | Test 01 | ✓ | ✓ | 0.00 | 0.00 |
| | Test 02 | ✓ | ✗ | 0.06 | 0.00 |
| | Test 03 | ✗ | ✓ | 0.43 | 0.00 |
| | Test 04 | ✗ | ✗ | 5.15 | 0.01 |
| | Average | - | - | 1.41 | **0.003** |

the weakest overall, reflecting its purely reconstructive pre-training signal. Aggregating across layers consistently helps: the attentive probe yields the best results for all models, with AudioMosaic reaching 33.5 mAP, a 3.3-point gain over its best single layer demonstrating that different layers capture complementary information. Overall, AudioMosaic performs strongly under both fine-tuning and linear probing, indicating that it learns more generalizable and transferable representations.

### 5.3. Deepfake Detection

Audio SSL encoders are widely used for audio deepfake detection due to their strong generalization ability, either by attaching a linear classifier or by adopting specialized

architectures such as AASIST (Jung et al., 2022). In this subsection, we evaluate the generalization of audio encoders on the Environmental Sound Deepfake Detection (EnvSDD) benchmark (Yin et al., 2025b), focusing on robustness to unseen data sources and unseen generative models.

We use both linear head and AASITI on top of the AudioMosaic encoder. We follow the same fine-tuning procedure as in Section 5.1 and evaluate using the official EnvSDD protocol. Performance is reported using equal error rate (EER), where lower values indicate better detection (Table 3).

AudioMosaic consistently achieves lower EER than the baselines across both unseen data sources and unseen generative models, for both text-to-audio (TTA) and audio-to-audio (ATA) deepfakes. Interestingly, AASIST, despite introducing additional parameters, does not provide a clear performance gain over a linear head. We believe that this is because performance on EnvSDD is already close to saturation, leaving limited room for further improvement. These results indicate that the representations learned by AudioMosaic generalize well beyond the pre-training distribution and are effective for detecting novel generative artifacts without relying on specialized detection architectures.

## 5.4. Audio-Language Models

Audio encoders can enhance the audio perception capabilities of multi-modal LLMs by enabling reasoning over audio content. This is typically achieved by aligning audio representations with the LLM embedding space through a projection layer. In this subsection, we follow the experimental setting of LTU (Gong et al., 2024), which uses curriculum learning over audio classification, description, and closed- and open-ended question answering tasks on the OpenAQA dataset. We replace the original CAV-MAE encoder (Gong et al., 2023) with our AudioMosaic encoder and align it with LLaMA-7B (Touvron et al., 2023), keeping all other settings identical. Note that the evaluation protocol in this subsection differs from those used in previous subsections, but follows the LTU setup.

During evaluation, the audio–LLM is prompted with "*Write an audio caption describing the sound.*" For classification tasks, model outputs are encoded into text embeddings and compared with class-label prompts released by LTU, using the same text encoder as in the original work (e.g., `gpt-text-embedding-ada` from OpenAI). Captioning performance is evaluated using the SPICE metric, and results are reported in Table 4.

Overall, replacing the original encoder with AudioMosaic improves performance on most audio–language tasks, particularly in the zero-shot setting, with only a minor decrease on DCASE. Qualitative examples in Appendix B further suggest that AudioMosaic enables the model to capture finer-grained acoustic details that are missed by the original encoder and, in some cases, by the reference captions. Together, these results indicate that AudioMosaic provides richer audio representations for audio–LLMs.

## 5.5. Ablations

In this subsection, we present ablation studies on masking strategies, mask ratios, and pre-training batch size. All results are based on fine-tuning on AS-20K, with a default batch size of 2048 unless otherwise specified.

**Impact of different masking strategies.** Figure 4a compares the proposed time–frequency masking strategy for constructing positive pairs with time-only, frequency-only, and unstructured masking under different mask ratios. Time-only and frequency-only indicate that positive pairs are constructed by masking only along the time or frequency dimension, respectively. Frequency-only and unstructured masking perform noticeably worse, while time-only masking is more competitive at low mask ratios, suggesting that temporal masking plays a primary role. However, at higher mask ratios, incorporating additional frequency masking further improves performance. This indicates that constructing positive pairs using structured time–frequency masking is most effective for contrastive learning on spectrograms.

**Impact of different masking ratio.** Figure 4b provides a more fine-grained analysis of different masking ratios. It shows that higher masking ratios along the time dimension are generally more beneficial, and that adding frequency masking on top of strong temporal masking further improves performance. These results suggest that temporal information is often more redundant, whereas frequency components may carry more discriminative cues, such as timbre and pitch. Strong time masking encourages the model to learn more global and invariant representations, while excessive frequency masking may remove important identity-related information. The best performance is achieved with $\rho_t = 0.6$ and $\rho_f = 0.4$.

**Impact of batch size and memory efficiency.** Figure 4c studies the effect of pre-training batch size. Larger batch sizes consistently improve contrastive learning performance, in line with prior findings in the literature (Chen et al., 2020c). For masking-based augmentation, we observe the same trend: performance continues to improve up to our default batch size of 6144, suggesting that even larger batch sizes could yield further gains.

Table 5 compares peak GPU memory usage during pre-training across different batch sizes. AudioMosaic, AudioMAE, and BEATs exhibit comparable memory footprints, scaling roughly linearly from ~3.5 GB at batch size 64 to ~25 GB at batch size 512. In contrast, EAT consumes 34.6 GB at batch size 64, which is more than $10\times$ higher

*Table 4.* Comparison of audio encoders aligned with LLaMA-7B (Touvron et al., 2023) under the same experimental setup as LTU (Gong et al., 2024). Only the audio encoder is replaced. † denotes the zero-shot setting, where the dataset is excluded from both pre-training and audio–LLM alignment. Best results are shown in **bold**.

| Method | Audio Encoder | Classification | | | | | | | | Captioning | |
|---|---|---|---|---|---|---|---|---|---|---|---|
| | | ESC-50 (Acc) | DCASE (Mi-F1) | VS (Acc) | TUT (Acc) | BJO (Acc) | VGG (Acc) | FSD (mAP) | AS (mAP) | AudioCaps (SPICE) | Clotho (SPICE) |
| LTU | CAV-MAE (Gong et al., 2023) | 82.0† | **50.5**† | 55.7† | 24.1† | 64.8† | 38.4 | 45.8 | 18.2 | 16.0 | 12.0 |
| Ours | AudioMosaic | **86.5**† | 48.9† | **68.2**† | **25.0**† | **66.1**† | **54.6** | **46.9** | **21.0** | **17.1** | **12.5** |

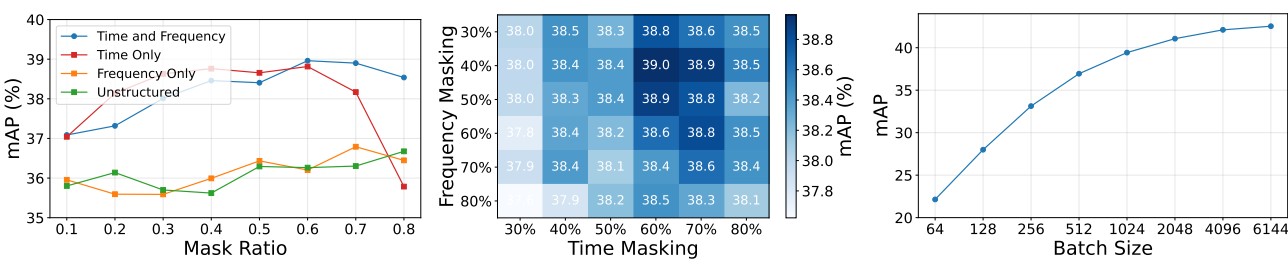

*(a)* Comparison of masking strategies.     *(b)* Ablation over mask ratio.     *(c)* Ablation over batch size.

*Figure 4.* (a) Comparison of masking strategies under different mask ratios. (b) Ablation on different mask ratios for time-frequency masking constructed positive pairs. (c) Ablation on pre-training batch size. All results are based on fine-tuning on AS-20K, with a default batch size of 2048.

*Table 5.* Peak GPU memory (GB) during pretraining with gradient checkpointing. All methods use a ViT-Base model on 10 s log-Mel spectrograms ($B \times 1024 \times 128$, where $B$ denotes the batch size). Memory usage is measured using `torch.cuda.max_memory_allocated()` on a single NVIDIA L40S GPU.

| Method / Batch Size | 64 | 128 | 256 | 512 |
|---|---|---|---|---|
| AudioMAE | 3.7 | 6.8 | 13.0 | 25.4 |
| BEATs | 3.6 | 6.7 | 13.1 | 25.9 |
| EAT | 34.6 | OOM | OOM | OOM |
| AudioMosaic (ours) | **3.3** | **6.3** | **12.3** | **24.3** |

than the other methods at the same batch size. This is primarily due to its `clone_batch=16` strategy, which expands each sample into 16 differently masked copies (effective batch size 1024), together with the need to maintain a full EMA teacher network in memory alongside the student encoder. As a result, EAT exceeds the 48 GB memory capacity of an L40s GPU at batch size 128, requiring either much smaller per-GPU batch sizes or substantially greater GPU memory. By contrast, AudioMosaic achieves competitive memory efficiency while still processing two augmented views per sample, since its structured time–frequency masking reduces each view to only ∼24% of the full token sequence.

Overall, the results in this subsection demonstrate the effectiveness of each component of the AudioMosaic encoder. Structured time–frequency masking is critical for effective contrastive learning on spectrograms, and stronger temporal

masking combined with moderate frequency masking yields the best performance.

## 6. Conclusion

In this work, we introduced AudioMosaic, a contrastively pre-trained audio encoder that constructs positive pairs through structured, independent time–frequency masking of spectrogram patches. By analyzing representation quality using effective rank, we showed that structured time–frequency masking encourages richer representations, leading to improved transferability across datasets, tasks, and acoustic conditions. We further demonstrated that aligning the pretrained AudioMosaic encoder with large language models improves performance on audio–language tasks, indicating that the learned representations are well suited for multimodal reasoning. Together, these results suggest that rethinking masking as a mechanism for contrastive view construction offers a practical and effective approach for learning general-purpose audio representations, with direct benefits for both standalone audio understanding and emerging audio–LLM systems.

## Acknowledgment

This research was conducted by the ARC Centre of Excellence for Automated Decision-Making and Society (CE200100005), and funded by the Australian Government through the Australian Research Council. This research was supported by The University of Melbourne's Research Computing Services and the Petascale Campus Initiative.

## Impact Statement

This work advances self-supervised audio representation learning and contributes to general-purpose audio understanding. The proposed method can benefit a range of downstream applications, including audio classification, audio–language modeling, and audio–LLM systems that require effective audio perception.

Improved audio representations may also support audio forensics tasks such as deepfake detection, helping mitigate misuse of generative audio technologies. As with many machine learning methods applied to audio data, responsible use and consideration of data privacy are important when deploying systems built upon this work.

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

## A. Experimental Settings

Table 6 summarizes the detailed experimental settings for both pre-training and fine-tuning, while Table 7 reports the settings used for linear probing. Table 8 further details the augmentations applied prior to spectrogram masking to generate two distinct views of the same audio clip. These settings largely follow prior work; we adjust only the learning rate to better suit fine-tuning the AudioMosaic encoder. All experiments are conducted with NVIDIA L40S GPUs (48GB).

*Table 6.* **Pre-training (PT) and Fine-tuning (FT) hyperparameters**. For augmentation, R: sampling random starting points with cyclic rolling in time; N: adding random noise (signal-to-noise ratio (SNR): 20dB) to spectrograms. For loss functions, BCE: binary cross entropy loss (for multi-label datasets or when using mixup (Zhang et al., 2018)); CE: cross-entropy loss, MSE: mean square error loss. * We used a fixed learning rate for pre-training.

| Configuration | Pre-training AS-2M PT | Fine-tuning | | | | | |
|---|---|---|---|---|---|---|---|
| | | AS-2M | AS-20K | ESC | SPC-2 | SPC-1 | EnvSDD |
| Optimizer | AdamW (Loshchilov & Hutter, 2019) | | | | | | |
| Optimizer momentum | $\beta_1 = 0.9, \beta_2 = 0.999$ | | | | | | |
| Weight decay | 0.01 | | | | | | |
| LR layer decay | 1.0 | 0.75 | 0.75 | 0.75 | 0.75 | 0.75 | 0.75 |
| Base learning rate | 0.0006* | 0.0002 | 0.0005 | 0.0005 | 0.0002 | 0.0002 | 0.001 |
| Learning rate schedule | Half-cycle cosine decay (Loshchilov & Hutter, 2017) | | | | | | |
| Minimum learning rate | 0.000001 | | | | | | |
| Gradient clipping | None | | | | | | |
| Warm-up epochs | 1 | 10 | 4 | 5 | 5 | 5 | 1 |
| Epochs | 400 | 100 | 60 | 60 | 200 | 100 | 60 |
| Batch size | 6144 | 256 | 64 | 16 | 64 | 64 | 128 |
| GPUs | 12 | 1 | 1 | 1 | 1 | 1 | 1 |
| Weighted sampling | False | True | False | False | False | False | False |
| Weighted sampling size | - | 200,000 | - | - | - | - | - |
| Augmentation | R | R | R | R | R+N | R+N | - |
| SpecAug (Park et al., 2019) (time/frequency) | - | 192/48 | 192/48 | 96/24 | 48/48 | 48/48 | 192/48 |
| Drop path (Huang et al., 2016) | 0.1 | 0.1 | 0.1 | 0.1 | 0.1 | 0.1 | 0.1 |
| Dropout (Srivastava et al., 2014) | 0.0 | 0.0 | 0.0 | 0.0 | 0.0 | 0.0 | 0.0 |
| Mixup (Zhang et al., 2018) | 0.0 | 0.8 | 0.8 | 0.0 | 0.8 | 0.8 | 0.0 |
| Multilabel | N/A | True | True | False | False | False | False |
| Loss Function | MSE | BCE | BCE | CE | BCE | BCE | CE |
| Dataset Mean for Normalization | -4.268 | -4.268 | -4.268 | -6.627 | -6.846 | -6.702 | -4.268 |
| Dataset Std for Normalization | 4.569 | 4.569 | 4.569 | 5.359 | 5.565 | 5.448 | 4.569 |

*Table 7.* **Linear probing hyperparameters**. All other settings are the same as the fine-tuning hyperparameters.

| Configuration | AS-2M | AS-20K | ESC | SPC-2 | SPC-1 |
|---|---|---|---|---|---|
| Learning rate | 0.1 | 0.001 | 0.1 | 0.1 | 0.1 |
| Epochs | 20 | 60 | 60 | 200 | 100 |
| Batch Size | 16 | 16 | 32 | 32 | 32 |

Table 9 presents an ablation study of different augmentation strategies. The results demonstrate that augmentation is necessary for constructing sufficiently distinct views; otherwise, even with masking, the model may still observe highly similar local structures across the two views. The results further show that time–frequency masking provides clear additional benefits beyond traditional contrastive learning based solely on waveform augmentations.

## B. Qualitative Examples

In this section, we present qualitative examples demonstrating that the AudioMosaic encoder improves the audio perception capabilities of LTU (Gong et al., 2024) on fine-grained details. In particular, AudioMosaic enables the model to capture details that are missed when using the original CAV-MAE encoder (Gong et al., 2023), as well as additional details that are not explicitly mentioned in the reference captions but have been verified by the authors.

*Table 8.* Augmentation strategies applied before spectrogram masking.

| Augmentation | Probability | Parameters |
|---|---|---|
| Polarity Inversion | 0.5 | - |
| Time Stretch | 0.7 | Minimum rate 0.7, Maximum Rate 1.25 |
| Gaussian Noise | 0.5 | Minimum SNR ratio 5.0 dB, Maximum SNR ratio 40 dB |
| Gain | 0.3 | Minimum gain -12 dB, Maximum gain 12dB |
| High Pass Filter | 0.3 | Minimum cutoff frequency in 20 Hz, Maximum cutoff frequency 2.4 kHz |
| BandStop Filter | 0.5 | Minimum center frequency 200 Hz, Maximum center frequency 4 kHz |
| PitchShift | 0.6 | Minimum -4 semitones to shift, Maximum 4 semitones to shift |

*Table 9.* Comparison of different waveform augmentation and masking strategies. Results are reported as fine-tuning performance on the AS-20K dataset.

| Augmentation | Masking | mAP |
|---|---|---|
| None | Time-frequency | 24.8 |
| + Polarity Inversion and Time Stretch | Time-frequency | 25.0 |
| +GaussianSNR + Gain | Time-frequency | 29.8 |
| +High Pass Filter +BandStop Filter | Time-frequency | 34.2 |
| Full Augmentation | Time-frequency | **42.5** |
| Full Augmentation | None | 30.1 |

---

**Audio clip from AudioSet (id: FR7BDRhMATo)**

**Prompt:** Close-ended question: Write an audio caption describing the sound

**Original LTU:** A train is moving and its horn blows.

**AudioMosaic:** A train whistle blows and a bell rings as it moves down the tracks.

**Reference Caption:** Blowing of a horn as a train passes.
**Note:** The bell ring can be heard at the end of the audio clip as the train moves down the tracks.

---

**Audio clip from AudioSet (id: wnqUgK_-fo4)**

**Prompt:** Close-ended question: Write an audio caption describing the sound

**Original LTU:** A crowd is cheering and whistling.

**AudioMosaic:** Whistling and cheering with fireworks.

**Reference Caption:** Firecrackers popping as a crowd of people cheer and whistle.

---

**Audio clip from AudioSet (id: wVi5w_NU6CM)**

**Prompt:** Close-ended question: Write an audio caption describing the sound

**Original LTU:** A fire engine siren is blaring nearby.

**AudioMosaic:** A fire engine siren is blaring loudly and a vehicle is driving by quickly.

**Reference Caption:** A fire engine horn blows, followed by a fire engine siren blowing.
**Note:** A vehicle driving by can be heard in the background.

