# OpenReview forum: "AudioMosaic: Contrastive Masked Audio Representation Learning"
_ICML.cc/2026/Conference — ICML 2026 regular_

### Official Review · Reviewer_bU2G · 2026-02-24

**Soundness:** 3
**Presentation:** 4
**Significance:** 3
**Originality:** 2
**Overall Recommendation:** 5
**Confidence:** 4

**Summary:**

This paper introduces AudioMosaic, a self-supervised learning framework for audio that combines contrastive learning and spectrogram masking. This addresses a gap in the literature, as most unsupervised learning-based feature extractors are time-domain based. The paper compares two masking strategies in the frequency domain: unstructured masking and time-frequency masking. They show that time-frequency masking performs representations more suitable for downstream tasks than the former using contrastive loss. Compared to contemporary unsupervised feature extractors, AudioMosaic demonstrate strong performance in downstream tasks, including deepfake detection, noise classification, and specific large language model integration.

**Compliance With Llm Reviewing Policy:**

Affirmed.

**Key Questions For Authors:**

Q1: How does the masking procedure work exactly? Am I interpreting it correctly that 'timebands' are drawn uniformly and removed for one view, and 'frequencybands' are also drawn uniformly and removed for another view?

Q2: To my understanding, the layer used for downstream tasks varies across domains (see e.g., Selection of Layers from Self-supervised Learning Models for Predicting
Mean-Opinion-Score of Speech, X. Liang, 2025, ASRU). How is the layer selected for the downstream experiments?

Q3: Which layer of the 12-layer ViT-B/16 is used to extract representations for the effective rank analysis? Does the conclusion hold across layers?

Q4: Related to the above question: Why is masking applied at inference time in the effective rank analysis? A standard use case would not do masking at the inference step. Or have I understood something wrong?

Q5: Table 7 lists seven augmentations applied before masking, but no ablation isolates their effect. How sensitive is the method to the specific augmentation choices? For example, would a simpler pipeline (e.g., two or three augmentations) suffice, or is the full seven-augmentation pipeline necessary for the reported performance?

**Limitations:**

Yes

**Strengths And Weaknesses:**

S1: The manuscript is well-written, and the argument for this study is clearly given in the introduction (gap in frequency-based representation learning). The approach is well illustrated (Fig. 1) and nicely presented.

S2: On significance, audio representation learning has a very large application domain, and using representations is the status quo for several audio fields today. This makes this work highly relevant. Several works have studied and benchmarked representation models for downstream tasks recently, providing support for the importance (see e.g., SHEET: A Multi-purpose Open-source Speech Human Evaluation Estimation Toolkit, W. Huang et. al., 2025, and Selection of Layers from Self-supervised Learning Models for Predicting Mean-Opinion-Score of Speech, X. Liang et. al., 2025).

S3: The approach seems sound; time and frequency masking on two views, respectively, trained using a contrastive loss, yield suitable representations for downstream tasks. The authors provide arguments for their design choices. The ablation study showcases why the authors chose particular hyperparameters.

S4: AudioMosaic report slight improvement over contemporary representation learning approaches. On the following datasets, measured using mean average precision (mAP)/accuracy in the downstream audio classification task, AudioMosaic achieve 42.5 mAP on AS-20K (second best, 40.9 mAP), 50.2 mAP on AS-2M (50.2 mAP), 97.5% on ESC-50 (96.3%), 98.4% on SPC-2 (best, 98.8%), and 99.0% on SPC-1 (second best, 98.8%). This suggests that the approach is highly effective.

W1: There is limited novelty in this work. Spectral masking is common in the speech field, and spectral masking for representation learning has been discussed in the representation learning community (e.g., Audio-MAE). The novelty in this work, to my knowledge, is applying time masking to one view, and frequency masking to another view, and applying contrastive loss over some network processing the views; this combination is non-obvious, and the manuscript provides empirical evidence that it works. However, the methodological novelty is incremental.

W2: The deepfake investigation is difficult to interpret. There, the authors compare representations used for deepfake prediction, but the representations of comparison uses AASIST prediction head, while AudioMosaic uses linear. To my understanding, the authors implicitly assume that AASIST is a superior prediction head over linear, but this is not supported. Using AASIST on AudioMosaic representations as well would strengthen the claim.

W3: Minor weak point, but the masking procedure is underspecified. The operators $M_t$ and $M_f$ are described as dropping "contiguous time regions" and "frequency bands," but the selection procedure is not defined; e.g., how many contiguous blocks are drawn, how are block sizes determined, and how are patches drawn (uniformly?)? Further, the variable $d$ on line 137 (second column) is undefined. These details are needed for reproducability.

---

> ### Author Rebuttal · Authors · 2026-03-31
>
> We are grateful for your careful review of our paper. Please find our detailed responses to your questions below.
>
> ---
>
> **W1:** Novelty and originality
>
> The contribution of this work is to rethink the role of masking in audio SSL. We demonstrate that time–frequency masking pairs are especially suitable for a contrastive objective, supported by both effective-rank analysis and comprehensive experimental results. We believe this combination is non-obvious unless demonstrated by this work. We also believe that providing an improved understanding is equally valuable as introducing a completely new methodology to the ICML community.
>
> ---
>
> **W2:** Using AASIST on AudioMosaic representations
>
> Please find below the additional results for AASIST using AudioMosaic representations. The results are similar to those obtained with a linear head and show a slight improvement over the ATA setting. We believe this may be because performance on this dataset is already close to saturation, leaving limited room for further gains.
>
> | Method \ Test Set | TTA-Test-01 | TTA-Test-02 | TTA-Test-03 | TTA-Test-04 | ATA-Test-01 | ATA-Test-02 | ATA-Test-03 | ATA-Test-04 |
> |:---:|:---:|:---:|:---:|:---:|:---:|:---:|:---:|:---:|
> | Wav2Vec 2.0 +AASIST | 0.26 | 13.04 | 10.60 | 45.80 | 0.38 | 26.59 | 13.30 | 52.40 |
> | BEATs +AASIST | 0.08 | 1.26 | 4.70 | 17.20 | 0.03 | 0.08 | 2.20 | 3.00 |
> | AudioMosaic +Linear (Ours) | **0.00** | **0.05** | **0.38** | **4.80** | **0.00** | **0.00** | 0.03 | 0.03 |
> | AudioMosaic +AASIST (Ours) | **0.00** | 0.06 | 0.43 | 5.15 | **0.00** | **0.00** | **0.00** | **0.01** |
>
> ---
>
> **W3/Q1:** Clarifications on the masking operators
>
> The masking operators $M_t(\cdot)$ and $M_f(\cdot)$ follow a block-wise masking strategy similar to prior works, where contiguous regions are sampled along the time or frequency axis. Specifically, masking blocks are sampled uniformly, and their sizes are determined to match the target masking ratios $\rho_t$ and $\rho_f$. Multiple blocks are applied until the desired proportion of patches is removed.
>
> We acknowledge that these details were not clearly specified in the current draft. We will revise the paper in the next version.
>
> ---
>
> **Q2:** Layer-wise selection for downstream tasks
>
> We use the final-layer representation from the Transformer blocks for downstream tasks, consistent with the baseline methods. We added an additional linear probing evaluation on AS-20K using different layers, weighted sums, and attention-based aggregation. The results show further improvements for the proposed method. We appreciate the reviewer’s suggestion and will incorporate this analysis in the next version.
>
> | Layer and method | 1 | 2 | 3 | 4 | 5 | 6 | 7 | 8 | 9 | 10 | 11 | 12 | Weighted Sum | Attention |
> |:---:|:---:|:---:|:---:|:---:|:---:|:---:|:---:|:---:|:---:|:---:|:---:|:---:|:---:|:---:|
> | mAP | 2.4 | 4.3 | 6.1 | 8.6 | 11.5 | 15.9 | 20.0 | 24.0 | 28.0 | 30.1 | 30.2 | 29.4 | 29.4 | 33.5 |
>
> ---
>
> **Q3/Q4:** Inference time effective rank and layer-wise evaluation
>
> We evaluated the no-masking setting at inference time and estimated its effective rank, see results below. Compared to the masked setting, the no-masking representations consistently show slightly lower effective rank. The overall trend remains consistent with our original findings and, therefore, does not affect any conclusions of the paper. We will update Figure 2 in the next revision to reflect this.
>
> We have also evaluated the layer-wise effective rank under the no-masking setting, with results shown in the table below. An interesting pattern emerges in the deeper layers, where the Time–Frequency masking pre-trained encoder consistently attains a higher effective rank. This is expected, since deeper representations are more influenced by the pre-training objective and may require higher intrinsic dimensionality to better distinguish different views.
>
> | Pre-training Method | No Mask | Unstructured | Time-Frequency |
> |:---:|:---:|:---:|:---:|
> | Unstructured | 5.09 | 5.22 | 5.30 |
> | Time-Frequency | 5.70 | 5.77 | 5.81 |
>
> | Pre-training Method \ Layers | 1 | 2 | 3 | 4 | 5 | 6 | 7 | 8 | 9 | 10 | 11 | 12 |
> |:---:|:---:|:---:|:---:|:---:|:---:|:---:|:---:|:---:|:---:|:---:|:---:|:---:|
> | Unstructured | 4.22 | 4.95 | 5.09 | 5.19 | 5.27 | 5.32 | 5.36 | 5.37 | 5.36 | 5.33 | 5.28 | 5.30 |
> | Time-Frequency | 4.57 | 4.98 | 5.10 | 5.17 | 5.26 | 5.38 | 5.49 | 5.69 | 5.73 | 5.82 | 5.84 | 5.81 |
>
> ---
>
> **Q5:** Ablation on augmentation
>
> The augmentation choices are largely heuristic, and we have added additional ablations to examine their impact, results are shown in the table below. We believe that exploring alternative augmentation strategies may further improve performance, and we will extend this analysis in the next version.
>
> | Augmentation | mAP |
> |---|---|
> | None | 28.6 |
> | +PolarityInv +TimeStretch | 28.8 |
> | +GaussianSNR +Gain | 34.2 |
> | +HighPass +BandStop | 39.4 |
> | +PitchShift | **42.5** |

---

> > ### Author Rebuttal · Reviewer_bU2G · 2026-04-01
> >
> > I thank the authors for the detailed response.
> >
> > W2: Thank you for including this. This shows that Linear is a surprisingly strong baseline, and that AASIST should not be considered as strictly more performant than Linear.
> >
> > W3/Q1: I still find this description too vague. Consider explaining it a bit more rigorously: consider a spectrogram $S\\in\\mathbb{R}^{T\times F}$. We sample a subset of $\\{1, 2, \\dots, F\\}$ and contiguous regions along the time dimensions to the sampled subset are then blocked... If this is correct.
> >
> > Q2: Thank you for doing this experiment. I find these results surprising and exciting, as there are other works suggesting that early layers are more appropriate for downstream tasks (e.g., in non-intrusive speech quality assessment). I am not well-versed in the audio classification literature, but it might be that later layers are more appropriate for the audio classification task. Also, for this downstream task, this layer analysis suggests that the last layer suffices to compare the suitability of the representations.
> >
> > Q5: Thank you for including this. This clearly shows the importance of the augmentations in the pipeline, and I think this ablation deserves a place in the main paper to help readers understand how much performance is attributable to augmentation versus masking strategy.
> >
> > I have also read reviewer Lwds's response, who gave (2) Reject as recommendation, and I acknowledge several of the points made, particularly on layer-analysis for downstream tasks and the coupled performance gains of the augmentations and the masking. In my assessment, the rebuttal contains new experiments that answer these questions, and I find that the remaining issues (framing of batch size claims, augmentation discussion) can be addressed in a revision without undermining the core contribution. The layer-analysis can also be interesting to include, but could be considered future work to experiment with several downstream tasks.
> >
> > I appreciate the work by the authors, and I am satisfied with the response (although W3/Q1 is not fully addressed, but this is minor). I will increase my score to Accept (5).

---

> > > ### Author Response · Authors · 2026-04-01
> > >
> > > Thank you for your kind recognition of our work. We will incorporate mathematical notation to better explain the spectrogram masking in the revised version.

---

### Official Review · Reviewer_hDpy · 2026-03-10

**Soundness:** 3
**Presentation:** 3
**Significance:** 3
**Originality:** 3
**Overall Recommendation:** 5
**Confidence:** 5

**Summary:**

This manuscript proposes AudioMosaic, a contrastive audio self-supervised learning framework that builds positive pairs by applying structured time–frequency masking to spectrogram patches, instead of following the now-dominant reconstruction-based masked modeling paradigm. Overall, the paper is strong, well motivated, and technically clean. Its central idea is simple but effective: rethinking masking not as corruption for reconstruction, but as a way to create complementary nontrivial views for contrastive learning. The paper also provides an effective-rank-based analysis to explain why the proposed masking strategy is better suited to contrastive learning on spectrograms.

**Compliance With Llm Reviewing Policy:**

Affirmed.

**Key Questions For Authors:**

The empirical results are strong, but the paper would benefit from a sharper articulation of why the proposed time–frequency masking strategy constitutes a substantial methodological advance rather than an effective engineering refinement.

The method appears to benefit from very large batch sizes. Can the authors provide more discussion or experiments on performance under smaller and more practically accessible training regimes, to clarify reproducibility for groups with limited compute?

Effective rank is a useful indicator, but it remains an indirect proxy. Are there additional analyses that could more directly connect the proposed masking strategy to improved invariance, reduced redundancy, or better semantic structure in the learned representations?

The paper includes diverse downstream tasks, which is a strength, but could the authors further clarify whether the proposed masking strategy is expected to transfer equally well across different spectrogram parameterizations, audio domains, or multimodal pretraining pipelines?

**Limitations:**

yes

**Strengths And Weaknesses:**

The main strengths are as follows. First, the method is conceptually elegant: the proposed time–frequency masking strategy is easy to understand and has a clear intuition, namely reducing shared local redundancy between positive views so that the model learns more global and discriminative utterance-level representations. This is a nice contribution because it addresses a real gap in spectrogram-based contrastive learning. Second, the paper offers more than empirical gains: the effective rank analysis gives a useful representation-level perspective, and the results suggest that AudioMosaic indeed learns less collapsed and more expressive features than reconstruction-based baselines. Third, the experimental results are strong and broad. The model performs very well on standard downstream benchmarks under both fine-tuning and linear probing, which is especially important because linear probing provides a cleaner test of representation quality. AudioMosaic also generalizes well to environmental sound deepfake detection and improves audio–language alignment performance when substituted into an LTU-style pipeline. Finally, the method is computationally attractive, since it avoids the decoder used in masked reconstruction and benefits from reduced visible-token counts under masking.

I also have several concerns.
First, while the paper is convincing empirically, the novelty is moderate rather than radical. The main contribution is a new masking strategy within a familiar contrastive learning pipeline, rather than a fundamentally new learning objective or architecture.
Second, the theoretical analysis remains limited. Effective rank is a useful diagnostic, but it is still only a proxy for representation quality. The analysis is suggestive rather than definitive, and the causal link between masking structure, effective rank, and downstream transfer is not fully established. A stronger theoretical treatment would make the paper more compelling.
Third, some experimental claims could be better stress-tested. Many baseline numbers are taken from original papers rather than re-run under a unified setting, which is common but still leaves room for confounding. Also, the strongest positive result in audio-language alignment is obtained by replacing the encoder in one existing LTU pipeline, which is useful but not a fully independent multimodal evaluation.
Fourth, the method appears to benefit from very large batch sizes, with the default pretraining batch size reaching 6144 and performance still increasing as batch size grows. Although the paper argues that the method is efficient, this dependence may limit accessibility and reproducibility for smaller-scale labs.

---

> ### Author Rebuttal · Authors · 2026-03-31
>
> We sincerely appreciate your review, valuable feedback, and kind recognition of our work. Below are our responses to your questions.
>
> ---
>
> **W1/Q1:** Novelty and originality
>
> The contribution of this work is to rethink the role of masking in audio SSL. We demonstrate that time–frequency masking pairs are especially suitable for a contrastive objective supported by comprehensive experimental results. We believe this combination is non-obvious unless demonstrated by this work. We also believe that providing an improved understanding is equally valuable as introducing a completely new methodology.
>
> ---
>
> **W2/Q1/Q3:** Theoretical analysis and effective rank
>
> We agree that effective rank is only an indirect proxy and does not by itself establish a causal link between masking structure and downstream transfer. However, prior works have demonstrated a strong correlation between effective rank, intrinsic dimension, and downstream transfer performance [1, 2]. In this self-supervised learning context, we believe it is a well-motivated and informative diagnostic for studying how different masking strategies and pre-training objectives relate to the quality of the learned representations.
>
> ---
>
> **W3:** Baseline results copied from original paper.
>
> We would like to clarify that only the results in Table 1 are taken from the original papers. All other reported results (including linear probing and comparisons with LTU) are obtained using publicly available pre-trained weights and evaluated with our own codebase and compute environment. For the comparison with LTU, we follow the original recipe and train all four stages, replacing only the encoder with our pre-trained model.
>
> For the fine-tuning results in Table 1, we likewise used publicly available pre-trained weights and attempted to reproduce the reported numbers. Our reproduced results on AudioSet are slightly lower than the originally reported values, likely because we were unable to access the full AudioSet release and therefore trained on fewer samples. To avoid unfair or confounded comparisons due to incomplete data, we report the original published numbers for the baseline methods in Table 1.
>
> ---
>
> **W4/Q2:** Batch size and reproducibility for groups with limited compute
>
> In our experiments on NVIDIA L40S GPUs (48GB), our method fits a batch size of 512 on a single GPU, with actual memory usage of approximately 24GB. We believe this represents a practical setting for relatively resource-constrained environments. In contrast, EAT reports a batch size of 12 across 4 RTX 3090 GPUs, totalling 96GB of GPU memory. The results below show that EAT requires roughly 10× more memory than the proposed method. The original submission already included an ablation over batch sizes down to 512. We further evaluated smaller batch sizes and found that, although performance decreases (33.1 / 28.0 / 22.1 mAP for batch sizes 256 / 128 / 64, respectively), it does not collapse or fail.
>
> | Method \ Batch Size | 64 | 128 | 256 | 512 |
> |:---:|:---:|:---:|:---:|:---:|
> | AudioMAE (80% Masking) | 3740 MB | 6925 MB | 13302 MB | 26050 MB |
> | BEATs | 3645 MB | 6911 MB | 13444 MB | 26507 MB |
> | EAT (teacher+student) | 35472 MB | OOM | OOM | OOM |
> | AudioMosaic (Ours) | **3418 MB** | **6481 MB** | **12600 MB** | **24837 MB** |
>
> We also note that additional techniques can be applied on top of our method. For example, gradient accumulation, as used in OpenCLIP, can further alleviate batch size constraints and enable contrastive pre-training under limited hardware resources. Therefore, we believe that the requirement for large batch sizes is not a fundamental limitation, even in resource-constrained environments.
>
> ---
>
> **Q4:** Transfer to different settings
>
> We believe the experimental coverage is already comprehensive. AudioSet spans major audio domains, including music, speech, and a wide range of general audio events, and our downstream evaluations likewise cover diverse sound categories. We further show that the proposed pre-trained encoder improves performance in audio-LLM settings, suggesting that the benefits of our method could generalize to broader use cases. For different spectrogram parameterizations, we follow the same setup as the baseline methods to ensure a fair comparison, which we believe is the appropriate choice and generalizable. We will include ablation over different spectrogram parameterizations in the next revision.
>
> ---
>
> [1] Dubois, Yann, Tatsunori Hashimoto, and Percy Liang. "Evaluating self-supervised learning via risk decomposition." In International Conference on Machine Learning, pp. 8779-8820. PMLR, 2023.
>
> [2] Garrido, Quentin, Randall Balestriero, Laurent Najman, and Yann Lecun. "Rankme: Assessing the downstream performance of pretrained self-supervised representations by their rank." In International conference on machine learning, pp. 10929-10974. PMLR, 2023.

---

> > ### Author Rebuttal · Reviewer_hDpy · 2026-04-03
> >
> > My concerns have been adddressed.

---

> > > ### Author Response · Authors · 2026-04-04
> > >
> > > We thank the reviewer for the positive feedback on our work. We will incorporate these clarifications in the revision.

---

### Official Review · Reviewer_Lwds · 2026-03-11

**Soundness:** 3
**Presentation:** 3
**Significance:** 3
**Originality:** 2
**Overall Recommendation:** 4
**Confidence:** 4

**Summary:**

AudioMosaic introduces a novel contrastive SSL method for audio. Unlike dominant generative pretext tasks that reconstruct masked spectrograms, AudioMosaic creates positive pairs for contrastive pre-training by applying independent time-frequency masking to spectrogram patches. This structured masking encourages the model to learn robust representations. As a result, AudioMosaic achieves state-of-the-art performance across various downstream audio and speech tasks.

**Compliance With Llm Reviewing Policy:**

Affirmed.

**Final Justification:**

As I mentioned in my rebuttal response, the paper in its original form had some major weaknesses that needed to be addressed. However, I think the authors did a good job tackling these issues within the short rebuttal period, particularly with the new augmentation ablations, the GPU efficiency vs. batch size comparison, and the exploration of different layers/probing techniques.

While it is hard to guarantee exactly how well and to what extent all these updates will be integrated into the final camera-ready version, I am willing to give the authors the benefit of the doubt. I recommend a weak accept, as the paper ultimately provides a very useful baseline for future Audio SSL research.

**Key Questions For Authors:**

I am open to raising my score, provided the authors can adequately address the weaknesses outlined above. The most important questions that would influence my final evaluation are:

- The introduction identifies the need for "large batch sizes" as a primary obstacle in contrastive SSL, yet your method relies on a massive batch size of 6144. At what lower-bound batch size does the contrastive objective actually collapse? Furthermore, how does your overall training efficiency (e.g., total compute, steps) compare to recent baselines like EAT that utilize lightweight CNN decoders?

- Table 7 outlines an extensive set of augmentations applied before masking. Can you provide an ablation study isolating the impact of these specific augmentations? It is crucial to decouple how much performance is driven by the novel time-frequency masking versus these standard augmentations. How does the model perform without these augmentations? How does the model perform without your masking approach? (see effective rank vs. performance) Do previous contrastive methods really "fail"? There is no comparison to them to see what really works

- What specific search ranges and processes were used for the downstream hyperparameters? Did you utilize layer-wise learning rate decay during fine-tuning (a standard practice for models like EAT)? Finally, can you provide evaluation results across multiple random seeds to establish statistical robustness?

**Limitations:**

yes

**Strengths And Weaknesses:**

## **Strengths**
- **Effective SSL approach**: The paper introduces a simple, yet effective, contrastive learning approach for audio SSL. By using independent time-frequency masking of spectrogram patches to construct positive pairs, it is a welcome addition to the masked-reconstruction dominated audio SSL landscape.
- **Strong results**: The papers claims to achieve state-of-the-art results/highly competitive performance across the general known evaluation datasets for audio SSL.
- **Addition of probing**: The inclusion of linear probing is highly appreciated, providing a much clearer picture of the learned representation quality than fine-tuning alone. This is a good contrast to conventional AudioSet SSL.
- **Multimodal integration**: Integrating the AudioMosaic encoder into an audio-language model to demonstrate its utility for audio-LLMs is a nice addition  to the conventional evaluation.
- **Writing and clarity:** The paper is very well-written with a clear narrative and logical storyline

## **Weaknesses**
While the paper presents a simple and effective contrastive learning approach for audio SSL with strong empirical results, there are gaps in the experimental design and ablation studies. Specifically, the paper’s claims regarding batch size efficiency contradict its own methodology, the impact of augmentations for SSL/pairs is left entirely unexplored. This raises the question whether the technique proposed here is the key advantage or does not. Furthermore, the efficiency claims are overstated relative to recent baselines, and hyperparameter optimization details are omitted.

### **Soundness & Experimental Design**
- **Batch size contradiction & limits:** In the introduction (and abstract), the authors state that contrastive pre-training models "typically rely on large batch sizes to provide sufficient negative samples" as a fundamental challenge/problem to solve and not to apply existing methods. However, their pre-training uses a very large batch size of 6144, which directly contradicts the motivation. While the ablation study also reduces to a batch size of 512, 512 is still quite large for resource-constrained environments. The paper does not show the lower bound: at what batch size does the contrastive objective actually collapse or fail? While the reduced token count lowers memory (L.193-203), there is no detailed and direct comparison to recent models for the reader to properly evaluate this. For instance, how does the number of epochs compare to the number of steps detailed in previous works like EAT [1]? What is the true efficiency advantage here? Can they be fairly compared?
- **Missing ablation for SSL augmentations:** This is a very significant weakness. Table 7 introduces a wide range of augmentations applied before spectrogram masking (Polarity Inversion, Time Stretch, Gaussian Noise, PitchShift, etc.) that are claimed to be "essential" in L.128. However, there is no ablation measuring the impact of these augmentations. It is currently impossible to decouple how much of the performance gain is driven by the independent time-frequency masking versus this extensive standard? augmentation pipeline (that also seems to be arbitrary). Do previous contrastive methods really fail? (see the following point) Where is the evidence for that claim? There is no comparison available. This is a critical missing ablation/investigation for a contrastive learning paper.
- **Missing correlation effective rank vs. performance**: The authors claim that effective rank correlates with downstream performance. While this is backed up by related work, it would be insightful if e.g., the unstructured CLR is also tested on the downstream tasks.
- **Unclear hyperparameter optimization:** The determination of hyperparameters for downstream evaluation also seems arbitrary. What was the hyperparameter optimization process? What ranges were searched? Additionally, the paper does not mention whether layer-wise learning rate decay was utilized during fine-tuning. This seems to be a standard practice for EAT, etc. While these papers also "hide" this information in the paper, it is very important to add it since it has a notable impact on fine-tuning performance.
- **Seeds for evaluation:** Although all previous audio SSL models do not report results across multiple random seeds, doing so is highly recommended. Providing only a single result for probing or fine-tuning lacks statistical robustness. Reporting the performance across multiple seeds would strengthen the validity of findings.
- **Linear probe and layer investigation**: While it's a very welcome addition to add linear probes to the evaluation, it relies solely on the last layer for linear probing. However, literature in audio benchmarking (e.g., SUPERB [2]) also extracts a weighted sum of all layers. Furthermore, recent work demonstrates that contrastive approaches dynamically shift different types of information to later layers compared to generative approaches [3]. An investigation into intermediate layer representations (or using other probing methods such as attentive probing) would notably improve the paper quality and provide a fair and complete assessment of the representation quality. This was also investigated in audio most recently in e.g., [4].

### **Originality & Significance**

- **Efficiency claims:** The authors claim an efficiency advantage over masked spectrogram modeling because prior works rely on "Transformer-based decoders". This argument is weak regarding current literature, as several recent masked-based audio SSL models (e.g., EAT [1]) utilize lightweight CNN decoders rather than Transformers. The efficiency claim should be contextualized against these more  lightweight generative baselines. Additionally, as mentioned above, the strong efficiency claims need to be better measured/evaluated/contextualized against the current work to better understand what the direct advantages are. They are not properly explained (or very vaguely).
- **Missing contextualization of evaluation claims:** In Section 5.1 (L.346-350), the authors claim that evaluations relying primarily on fine-tuning "may overestimate differences in representation quality". This is a good observation, but it is presented as a novel insight. This phenomenon has been studied in very recent audio [4] and vision [5] papers, which should be discussed here to strengthen this claim. Adding an "evaluation of (audio) ssl models" section to Related Work may also help this claim.
- **SSL method:**: While the simplicity of the contrastive method can be considered a strength of the paper, there is no evidence /no ablations that investigates the true impact of it. Especially the addition of additional augmentations without an investigation or showing how previous contrastive methods would fail is a significant weakness.

### **Presentation & Clarity**
- While generally very good, the presentation of the hyperparameter tables (Tables 5, 6, and 7) lacks text explaining why these specific configurations were chosen, leaving the methodology hard to reproduce or build upon the work.

[1] Chen et al.: EAT: Self-Supervised Pre-Training with Efficient Audio Transformer

[2] Yang et al.: SUPERB: Speech processing Universal PERformance Benchmark

[3] Alkin et al.: MIM-Refiner: A Contrastive Learning Boost from Intermediate Pre-Trained Representations

[4] Rauch et al.: Unmute the Patch Tokens: Rethinking Probing in Multi-Label Audio Classification

[5] Psomas et al.: Attention, Please! Revisiting Attentive Probing Through the Lens of Efficiency

---

> ### Author Rebuttal · Authors · 2026-03-31
>
> We sincerely appreciate your time and careful review of our work. Below, we provide detailed responses to each of your concerns. We will incorporate these clarifications, the suggested references and experiments in the revised paper.
>
> ---
>
> **W1/Q1** GPU efficiency claims and comparison with EAT.
>
> We would like to clarify that **the efficiency claim is not about eliminating the need for large batch sizes, but rather that our method is more memory-efficient during pre-training, thereby enabling larger batch sizes under the same hardware constraints.**
>
> In our experiments on NVIDIA L40S GPUs (48GB), our method fits a batch size of 512 on a single GPU, with actual memory usage of approximately 24GB. We believe this represents a practical setting for relatively resource-constrained environments. In contrast, EAT reports a batch size of 12 across 4 RTX 3090 GPUs, totalling 96GB of GPU memory. The results below show that EAT requires roughly 10× more memory than the proposed method. We also additionally experimented with smaller batch sizes and found that, although performance decreases (33.1 / 28.0 / 22.1 mAP for batch sizes 256 / 128 / 64, respectively), it does not collapse or fail.
>
> | Method \ Batch Size | 64 | 128 | 256 | 512 |
> |:---:|:---:|:---:|:---:|:---:|
> | AudioMAE (80% Masking) | 3740 MB | 6925 MB | 13302 MB | 26050 MB |
> | BEATs | 3645 MB | 6911 MB | 13444 MB | 26507 MB |
> | EAT (teacher+student) | 35472 MB | OOM | OOM | OOM |
> | AudioMosaic (Ours) | **3418 MB** | **6481 MB** | **12600 MB** | **24837 MB** |
>
> Regarding pre-training duration, EAT reports 400K pre-training steps, whereas we use 128K steps. Therefore, compared with EAT, our method uses less memory, supports substantially larger per-GPU batch sizes, and requires fewer pre-training steps.
>
> We also note that additional techniques can be applied on top of our method. For example, gradient accumulation, as used in OpenCLIP, can further alleviate batch size constraints and enable contrastive pre-training under limited hardware resources. Therefore, we believe that the requirement for large batch sizes is not a fundamental limitation, even in resource-constrained environments.
>
> ---
>
> **W2/Q2**: Ablation on augmentation
>
> We have added additional augmentation ablation results below (AS-20K fine-tuning). These ablations support our claim that augmentation is necessary for constructing sufficiently distinctive views; otherwise, even with masking, the model may still observe identical across the two views. These results also show that time–frequency masking provides clear additional benefits beyond a traditional contrastive approach based on augmentation alone.
>
> | Augmentation | Mask | mAP |
> |---|---|---|
> | Full Augmentation | None | 34.7 |
> | None | Time-frequency | 28.6 |
> | Full Augmentation | Time-frequency | **42.5** |
>
> We agree that the augmentation choices are somewhat heuristic, and we have added an additional ablation to examine this point. The results further support the discussion above.
>
> | Augmentation | mAP |
> |---|---|
> | None | 28.6 |
> | +PolarityInv +TimeStretch | 28.8 |
> | +GaussianSNR +Gain | 34.2 |
> | +HighPass +BandStop | 39.4 |
> | +PitchShift  | **42.5** |
>
> ---
>
> **W3:** Unstructured CLR performance
>
> We refer the reviewer to Figure 3(a), where the green curve already shows the fine-tuning performance of CLR with unstructured masking. In particular, the result at 80% masking, the best-performing setting for unstructured masking, corresponds to the result on effective rank shown in Figure 2.
>
> ---
>
> **W4/Q3:** What was the hyperparameter optimization process?
>
> All implementation details and most hyperparameters were inherited from the corresponding baseline codebases; thus, we also used the layer-wise decay for the learning rate decay. We did not perform hyperparameter tuning beyond the fine-tuning learning rate. For fine-tuning, rather than conducting a full grid search, we performed a small manual coarse search and selected values based on the observed training loss behavior.
>
> We agree that these details are important for reproducibility, especially since they are often omitted or only briefly mentioned in prior work. We will revise the paper to explicitly report these settings.
>
> ---
>
> **W5/Q3**: Evaluation under different seeds
>
> We have added additional fine-tuning results using three different random seeds (distinct from the original experiments). We do not observe any significant deviation from the originally reported evaluation results, with the largest standard deviation being 0.1.
>
> | AS-20k | ESC-50 | SPC-1 | SPC-2 |
> |:---:|:---:|:---:|:---:|
> | 42.5 ± 0.1 | 97.7 ± 0.0 | 99.1 ± 0.0 | 98.4 ± 0.0 |
>
> ---
>
> **W6**: Linear probe and layer investigation
>
> Please refer to the response to reviewer bU2G in Q2.

---

> > ### Author Rebuttal · Reviewer_Lwds · 2026-04-03
> >
> > Thank you for your thorough answer and the additional experiments in the short time. While not all of my concerns are addressed to a 100% (specifically more augmentation details, probing/embedding differences between your contrastive and generative models, and exactly how/to what extent these additions will be incorporated into the final paper) I still think this is a good contribution for audio SSL.
> >
> > Based on your responses and the consensus among the other reviewers, I will raise my score to a 4.

---

> > > ### Author Response · Authors · 2026-04-04
> > >
> > > We thank the reviewer for the positive feedback and for confirming that our response addressed the concerns.
> > >
> > > In the revision, we will extend the analysis by incorporating more detailed augmentation ablations and by applying layer-wise probing, weighted-sum probing, and attention-based linear probing to all baseline methods. We believe these additional results will provide further insight into audio SSL.

---

### Official Review · Reviewer_HSQP · 2026-03-13

**Soundness:** 3
**Presentation:** 3
**Significance:** 3
**Originality:** 2
**Overall Recommendation:** 4
**Confidence:** 2

**Summary:**

This paper proposes AudioMosaic, a self-supervised method for learning audio representations. The key idea is simple: instead of reconstructing masked audio, the model creates two views by masking spectrogram patches along the time and frequency axes, then trains them with contrastive learning. This reduces local overlap between the views, so the encoder has to capture more global information rather than relying on small local patterns. The paper also analyzes the learned representations using effective rank. The results suggest that the masking strategy helps avoid dimension collapse and produces higher-dimensional embeddings.
Experiments are conducted on datasets such as AudioSet and ESC-50. The method achieves strong performance under both fine-tuning and linear probing. The training pipeline is also relatively lightweight since masked tokens are dropped and the projection head is just an MLP. The pretrained encoder is further tested on tasks like deepfake detection and alignment with audio-language models.

**Compliance With Llm Reviewing Policy:**

Affirmed.

**Key Questions For Authors:**

The paper states that AudioMosaic is more efficient than generative approaches such as Audio-MAE. It would help to include concrete numbers, such as training time per epoch, peak GPU memory usage, or sample throughput.

In Section 5.5, how are the masking ratios chosen for the Time Only and Frequency Only strategies? Also, the evaluation tasks and metrics in this section should be clearly specified in the text or figure captions.

The results suggest that the combination 𝜌𝑡=0.6 and 𝜌𝑓=0.4 works best. From an audio perspective, why does stronger time masking with moderate frequency masking lead to better representations?

**Limitations:**

yes

**Strengths And Weaknesses:**

Strengths:

1. Solid experiments. The method is tested on multiple datasets, including AudioSet and ESC-50. Results for both fine-tuning and linear probing are competitive.

2. Simple and reasonable design. The structured masking over time and frequency is intuitive and helps reduce local redundancy between views.

3. Good analysis. The effective rank analysis helps explain why the representation does not collapse.

4. Good transfer ability. The learned features work well on other tasks, such as deepfake detection and audio-language alignment.

Weaknesses:

1. Limited novelty. The framework mostly follows a standard contrastive setup with an AST backbone. The main change is the masking pattern.

2. Efficiency is not quantified. The paper claims the method is efficient, but there are no concrete numbers such as training time, GPU memory usage, or throughput.

3. Some details are unclear. The ablation study in Section 5.5 is not very clear. For example, how the positive pairs are formed in the “Time Only” and “Frequency Only” settings is not fully explained. The evaluation tasks and metrics used in this section are also not clearly labeled.

---

> ### Author Rebuttal · Authors · 2026-03-31
>
> We sincerely appreciate your thorough review and insightful comments. Please find our responses to your questions below.
>
> ---
>
> **W1:** Novelty and Originality
>
> The contribution of this work is to rethink the role of masking in audio SSL. While previous works mainly use masking for spectrogram reconstruction, we show that it can also be used effectively for contrastive learning. In particular, we demonstrate that time–frequency masking pairs are especially suitable for a contrastive objective, supported by both effective-rank analysis and comprehensive experimental results. **We believe this combination is non-obvious unless demonstrated by this work. We also believe that providing an improved understanding is equally valuable as introducing a completely new methodology.**
>
> ---
>
> **W2/Q1:** Efficiency and GPU memory usage.
>
> In our experiments on NVIDIA L40S GPUs (48GB), our method fits a batch size of 512 on a single GPU, with actual memory usage of approximately 24GB. We believe this represents a practical setting for relatively resource-constrained environments. In contrast, EAT reports a batch size of 12 across 4 RTX 3090 GPUs, totalling 96GB of GPU memory. The results below show that EAT requires roughly 10× more memory than our proposed method. The training for AudioMosaic encoder takes ~20 hours for 128K pre-training steps, EAT (default 400K steps) estimated to cost ~90 hours on the same hardware.
>
> For clarity, we provide below a detailed comparison of pre-training memory usage against the baseline, and we will update the paper to make this point more explicit.
>
> | Method \ Batch Size | 64 | 128 | 256 | 512 |
> |:---:|:---:|:---:|:---:|:---:|
> | AudioMAE (80% Masking) | 3740 MB | 6925 MB | 13302 MB | 26050 MB |
> | BEATs | 3645 MB | 6911 MB | 13444 MB | 26507 MB |
> | EAT (teacher+student) | 35472 MB | OOM | OOM | OOM |
> | AudioMosaic (Ours) | **3418 MB** | **6481 MB** | **12600 MB** | **24837 MB** |
>
> We also note that additional techniques can be applied on top of our method. For example, gradient accumulation, as used in OpenCLIP and related works [1, 2], can further alleviate batch size constraints and enable contrastive pre-training under limited hardware resources. Therefore, we believe that the requirement for large batch sizes is not a fundamental limitation, even in resource-constrained environments.
>
> ---
>
> **W3/Q2:** “Time Only” and “Frequency Only” settings
>
> For the “Time Only” and “Frequency Only” settings, both views are randomly masked along the same dimension, either time only or frequency only. All other settings are exactly the same. This ensures a fair comparison with the proposed method.
>
> For the evaluation in Section 5.5, Figures 3(a) and 3(b) report fine-tuning performance on AS-20K using mAP as the evaluation metric. Each point corresponds to an individual pre-training run under a smaller-resource setting, using a batch size of 512 and 48K pre-training steps. Figure 3(c) uses the same evaluation protocol, except that it varies the batch size and uses 128K pre-training steps for fair comparison with the default setting. We will revise Section 5.5 and the captions of Figure 3 to make these details clearer.
>
> ---
>
> **Q3:** From an audio perspective, why does stronger time masking with moderate frequency masking lead to better representations?
>
> We believe temporal information is often more redundant, while frequency components carry more discriminative cues (e.g., timbre and pitch). Strong time masking encourages the model to learn more global and invariant representations, whereas excessive frequency masking can remove essential identity information. Our experimental results in Section 5 show that combining stronger time masking (60%) with moderate frequency masking (40%) provides a better balance between invariance and informativeness, leading to improved representations.
>
> ---
>
> [1] Cui, Q., Zhou, B., Guo, Y., Yin, W., Wu, H., Yoshie, O., & Chen, Y. (2022, October). Contrastive vision-language pre-training with limited resources. In European Conference on Computer Vision (pp. 236-253). Cham: Springer Nature Switzerland.
>
> [2] Pham, H., Dai, Z., Ghiasi, G., Kawaguchi, K., Liu, H., Yu, A. W., ... & Le, Q. V. (2023). Combined scaling for zero-shot transfer learning. Neurocomputing, 555, 126658.

---

> > ### Author Rebuttal · Reviewer_HSQP · 2026-04-03
> >
> > W1: I appreciate the authors' explanation regarding the non-obvious nature of combining time-frequency masking with contrastive learning. The effective-rank analysis provides good support for this. While the framework still leans on standard backbones, the insights are valuable.
> > W2/Q1: The authors have successfully addressed my concerns by providing a detailed quantitative comparison table.
> > W3/Q2 : The clarifications on the "Time Only" and "Frequency Only" settings, the metrics (mAP), and the dataset (AS-20K) are helpful. I look forward to seeing these details incorporated into the revised version.
> > Q3 : The explanation regarding temporal redundancy vs. frequency-domain discriminative cues is insightful and provides a solid scientific intuition for the chosen masking ratios.
> > Overall, the authors have addressed all my questions. I will maintain my score at 4 (Weak Accept) and recommend the paper for acceptance.

---

> > > ### Author Response · Authors · 2026-04-04
> > >
> > > We thank the reviewer for the positive feedback on our work and will incorporate these clarifications in the revision.

---

### Decision · Program_Chairs · 2026-04-30

**Decision:**

Accept (regular)

**Comment:**

Overview of the paper:
This paper introduces AudioMosaic, a contrastive self-supervised learning (SSL) framework for audio representation. Instead of utilizing the standard masked-reconstruction paradigm, the method generates positive pairs for contrastive learning by applying independent time-frequency masking to spectrogram patches. This approach is designed to reduce local redundancy between views, prompting the model to capture global, utterance-level representations without requiring generative decoders. The framework is evaluated on several downstream tasks, including standard audio classification benchmarks, environmental sound deepfake detection, and multimodal audio-language model integration.

Strengths:
- Methodological Clarity: Reviewers liked the straightforward and intuitive nature of the structured time-frequency masking strategy, finding it a sensible alternative to reconstruction-based audio SSL methods.
- Empirical Performance: The experimental evaluations cover multiple benchmarks and downstream applications. The method demonstrates competitive results under both fine-tuning and linear probing, as well as transferability to tasks such as deepfake detection and audio-language model alignment.
- Representation Analysis: Reviewers acknowledged the utility of the effective-rank analysis, which provides a quantitative perspective on how the masking strategy affects the dimensionality and expressiveness of the learned features.
- Computational Efficiency: By dropping masked tokens and omitting the decoder typically used in masked reconstruction, the framework reduces memory requirements and allows for larger per-GPU batch sizes compared to specific recent baselines.

Areas for improvement:
- Augmentation and Batch Size Dependencies: Initial reviews raised questions about the model's reliance on large batch sizes and a heuristic pre-masking augmentation pipeline. While the authors provided clarifying ablation studies and efficiency comparisons during the rebuttal, reviewers emphasized that these dependencies need careful contextualization in the final manuscript.
- Clarity of Implementation Details: The initial submission lacked specific details regarding the mathematical formulation of the block-masking operators, the selection of downstream layers for evaluation, and the hyperparameter optimization process. The authors have agreed to incorporate these specifications, along with additional ablation tables, in the camera-ready version.